# Oral Maintenance Therapy in Early Breast Cancer—How Many Patients Are Potential Candidates?

**DOI:** 10.3390/cancers17010145

**Published:** 2025-01-05

**Authors:** Nikolas Tauber, Lisbeth Hilmer, Dominik Dannehl, Franziska Fick, Franziska Hemptenmacher, Natalia Krawczyk, Thomas Meyer-Lehnert, Kay Milewski, Henriette Princk, Andreas Hartkopf, Achim Rody, Maggie Banys-Paluchowski

**Affiliations:** 1Department of Gynecology and Obstetrics, University Hospital Schleswig-Holstein, Campus Lübeck, 23538 Lübeck, Germany; lisbeth.hilmer@uksh.de (L.H.); franziska.fick@uksh.de (F.F.); franziska.hemptenmacher@uksh.de (F.H.); kay.milewski@uksh.de (K.M.); henriette.princk@uksh.de (H.P.); achim.rody@uksh.de (A.R.); maggie.banys-paluchowski@uksh.de (M.B.-P.); 2Department of Gynaecology and Obstetrics, University of Tuebingen, 72016 Tuebingen, Germany; dominik.dannehl@med.uni-tuebingen.de (D.D.); andreas.hartkopf@med.uni-tuebingen.de (A.H.); 3Department of Gynaecology and Obstetrics, University Hospital Duesseldorf, 40225 Duesseldorf, Germany; natalia.krawczyk@med.uni-duesseldorf.de; 4Department of Computer Science, ETH Zurich, 8092 Zurich, Switzerland; thmeyer@ethz.ch

**Keywords:** oral maintenance therapy, early breast cancer, hormone receptor positive, HER2 negative, combined endocrine therapy

## Abstract

Hormone receptor-positive and human epidermal growth factor receptor 2-negative (HR+/HER2−) early breast cancer (eBC) is the most common breast cancer subtype. While all HR+ patients receive adjuvant endocrine therapy, the addition of the PARP inhibitor olaparib or the CDK4/6 inhibitors abemaciclib or ribociclib is recommended only for intermediate- to high-risk carcinomas. This retrospective single-center analysis utilized patient data from the past 10 years. We aimed to determine how many patients in a real-world cohort met the criteria for endocrine-based combination therapy compared to those in the original approval study cohorts. A total of 146 patients (7.2%) met the eligibility criteria for olaparib, 312 patients (15.3%) fulfilled the criteria for abemaciclib, and 685 patients (33.6%) qualified for ribociclib. To our Knowledge, this is the largest analysis of its kind and it is crucial to estimate the potential increase in clinical workload following the approval of ribociclib to ensure the maintenance of high therapeutic standards.

## 1. Introduction

Worldwide, breast cancer caused 670,000 deaths in 2022, making it the leading cause of cancer-related deaths among women. Additionally, there were 8.2 million people alive who had been diagnosed with breast cancer in the past five years [1,2]. Approximately 70% of early breast cancer (eBC) cases are hormone receptor positive (HR+) and human epidermal growth factor receptor 2 negative (HER2−) [3,4,5]. Current therapy guidelines recommend that all patients with HR+ eBC receive either tamoxifen or aromatase inhibitors (AI), with or without ovarian suppression depending on individual risk and menopausal status, for at least five years in the adjuvant setting [6,7,8]. However, the risk of local or distant recurrence remains elevated particularly in patients with large node-positive tumors or an unfavorable tumor biology [9,10,11,12]. For risk-adapted (post-neo)adjuvant treatment, combining endocrine therapy with poly(adenosine diphosphate-ribose) polymerase (PARP) or cyclin-dependent kinase 4 and 6 (CDK4/6) inhibitors has been shown to significantly improve outcomes in intermediate to high-risk patients with HR+ HER2− disease [3,4,9,13,14,15,16]. Currently approved agents for endocrine-based maintenance therapy are olaparib, abemaciclib, and ribociclib. All three agents have been examined in randomized phase III trials; however, the inclusion criteria in each trial differed, resulting in different patient populations eligible for treatment (Table 1, Figure 1).

While the CDK4/6 inhibitors abemaciclib and ribociclib are indicated based on tumor characteristics, the PARP inhibitor olaparib is approved for high-risk patients with germline breast cancer (BRCA) 1/2 mutations. The randomized OlympiA trial [9,13] was originally designed for triple negative eBC but amended for HR+ HER2− tumors with high risk of recurrence. Eligible patients were required to have completed definitive local treatment, including radiotherapy, as well as (neo-)adjuvant chemotherapy, containing anthracyclines, taxanes, or a combination of both for at least six cycles. The use of platinum agents was allowed. To meet high-risk criteria, patients with HR+ HER2− eBC receiving primary surgery and adjuvant chemotherapy were required to have at least four pathologically confirmed metastatic lymph nodes. Patients receiving neoadjuvant chemotherapy (NACT) were eligible in cases of residual disease at the time of surgery (no pathologic complete response = non-pCR) and a “Combined Positive Score for Estrogen and HER2 receptors” (CPS + EG score) ≥ 3. The CPS + EG score is calculated based on the clinical and pathological stage as well as the estrogen receptor status and nuclear grade and correlated with relapse probability after neoadjuvant chemotherapy [13,17]. Olaparib was administered for one year and combined with adjuvant endocrine therapy.

Unlike PARP inhibitors, CDK4/6 inhibitors provide therapeutic potential for a broader patient population, as their effectiveness is independent of BRCA mutation status. Three CDK4/6 inhibitors are approved for use in the metastatic setting. However, only two have demonstrated a significant benefit in eBC, whereas palbociclib failed to do so in the PENELOPE-B and PALLAS trials [14,18,19]. Abemaciclib was approved for eBC following the randomized monarchE study [3,14]. The trial focused on patients with high-risk disease characteristics, such as high nodal burden (≥4 positive nodes) or 1–3 positive nodes and at least one of the following: (cohort 1) tumor size ≥ 5 cm, histologic grade 3, or (cohort 2) centrally assessed Ki-67 ≥ 20%. However, the approval label does not include cohort 2. In contrast to the OlympiA and monarchE trials, the NATALEE study investigating ribociclib also included patients with intermediate risk of relapse [4,15,16]. Eligibility required patients either to be node-positive or to have stage IIB/III eBC at the worst point since diagnosis. Additionally, patients with stage IIA node-negative eBC were included if they met one of the following criteria: histologic grade G3, or G2 with either a high proliferation index (Ki-67 ≥ 20%) or classification in a genomic high-risk group. In total, 28% of the study participants had node-negative disease at the time of diagnosis.

While all three agents improved outcomes in selected subgroups of HR+ HER2− eBC patients, real-world experience is still limited. Notably, the application of the NATALEE inclusion criteria is expected to significantly increase patient volume [20]. The aim of the present analysis was to evaluate how many patients can be considered potential candidates for one or more maintenance therapy options at a large university breast cancer center.

## 2. Materials and Methods

The single-center retrospective analysis was conducted at the certified Breast Cancer Center of the University Hospital Schleswig-Holstein, Campus Lübeck, and included all patients who were treated for eBC between January 2014 and December 2023. This analysis was carried out in accordance with the guidelines of the Declaration of Helsinki and approved by the Ethical Committee of the University of Lübeck (file number: 2024-496). The analysis included both men and women with non-metastatic disease. The evaluation of HR and HER2 receptor expression was performed by certified pathologists in accordance with established local standards. Tumors were classified as HR+ if they exhibited positive ER and/or PR expression via immunohistochemistry, with a minimum of 1% for ER and 1% for PR. HER2 immunoreactivity was assessed on a scale of 0 to 3+ using the approved test. Tumors with a immunohistochemical score of 2+ were analyzed by in situ hybridization. For the assessment of HER2 status, ASCO/CAP guidelines at the time of tissue examination were followed.

In order to identify the potential candidates for the respective substances olaparib, abemaciclib, and ribociclib, the approval criteria for these substances were applied. In patients who met the OlympiA inclusion criteria, the gBRCA1/2 mutation status was not systematically recorded due to the retrospective nature of the analysis. According to the OlympiA trial CPS+EG scores were calculated for potential candidates of olaparib using the “Neoadjuvant Therapy Outcomes Calculator” from the MD Anderson Cancer Center, University of Texas, based on the required parameters (https://www3.mdanderson.org/app/medcalc/index.cfm?pagename=bcnt [accessed 4 January 2025). In line with the monarchE and NATALEE studies, the indication for abemaciclib and ribociclib for patients undergoing neoadjuvant chemotherapy was based on the pretherapeutic or postoperative tumor (T), node (N), and metastasis (M) (TNM) stage, whichever was prognostically worse.

The data analysis was conducted using Excel 2311 and Statistical Package for Social Sciences (IBM SPSS Statistics, Version 29.0.2.0, Armonk, NY, USA: IBM Corp). The results were used to create a Venn diagram using Python 3.10.12 with the following libraries: math, matplotlib.pyplot, and matplotlib_venn.

## 3. Results

A total of 3230 patients with newly diagnosed breast cancer were identified (Figure 2). A total of 268 patients with ductal carcinoma in situ (DCIS) and 293 with de novo metastatic disease were excluded. The most prevalent tumor subtype among patients with invasive eBC was HR+ HER2− (2038, 63.1%), followed by HER2+ (10.5%) and TNBC (9.0%). In total, 2038 patients with invasive HR+ HER2− eBC were included in further analyses. Out of these patients, 244 (12.0%) received NACT and 1794 (88.0%) underwent primary surgical treatment.

### 3.1. Olaparib

In total, 146 out of 2038 patients (7.2%) would have met the criteria for potential use of olaparib according to the OlympiA trial, provided they had a germline BRCA1 or BRCA2 mutation. Due to the retrospective nature of our study, the data on the prevalence of (likely) pathogenic mutations in this patient population were not available. Among the 1794 patients undergoing primary surgery, 60 (3.3%) had a pN2 stage (4 to 9 positive axillary lymph nodes) and 35 (2.0%) pN3 stage (≥10 positive lymph nodes). Out of the 244 patients who received NACT, 51 (20.9%) had non-pCR and a CPS + EG score ≥ 3 (39 [16.0%] CPS + EG score 3 and 12 [4.91%] score 4). In the case of a gBRCA mutation, these patients would have been eligible for olaparib therapy.

### 3.2. Abemaciclib

The criteria for abemaciclib according to the approval label (i.e., cohort 1 of the monarchE study) were met by a total of 312 (15.3%) out of the 2038 patients. Among these, 95 (30.4%) had ≥4 positive lymph nodes after primary surgery, and 36 (11.5%) had ≥4 positive/suspicious axillary lymph nodes before and/or after NACT (≥cN2 and/or ≥ypN2). Furthermore, 48 (15.4%) had a tumor size of ≥5 cm with 1–3 positive lymph nodes after primary surgery, while 25 (8.0%) had tumors ≥ 5 cm and 1–3 positive/suspicious axillary lymph nodes before and/or after NACT (cN1 and/or ypN1). A total of 68 (21.8%) patients had grade 3 tumors < 5 cm and 1–3 positive lymph nodes after primary surgery, while 40 (12.8%) had grade 3 tumors < 5 cm and 1–3 positive/suspicious nodes before and/or after NACT (cN1/ypN1). As mentioned before, the approval criteria do not include cohort 2 (i.e., patients with 1–3 positive nodes and Ki67 ≥ 20%). If these criteria had been included in the approval label, an additional 46 patients (12.9%) would have been eligible for abemaciclib, while for 63 additional patients with 1–3 positive/suspicious axillary lymph nodes, no Ki-67 value was available.

### 3.3. Ribociclib

A total of 685 patients (33.6%) met the indication criteria for ribociclib according to the NATALEE study. Within this cohort, 523 (76.4%) had node-positive disease, thereby meeting the indication for ribociclib based solely on nodal status. Among the 162 node-negative patients, 47 (29.0%) had a T3/T4 stage, and 76 (46.9%) had a T3/G2 stage. In addition, 33 (20.4%) patients met NATALEE study criteria due to a T2/G2 stage with a Ki67 index of ≥20%, and only 6 (3.7%) due to a T2/G2 stage with an additional high genomic risk (e.g., Oncotype DX with a recurrence score ≥ 26 or high-risk profile in Prosigna/PAM50, Mammaprint, or Endopredict). It should be noted, however, that 47 patients with a T2/G2 stage and low Ki67 index did not undergo genomic testing and therefore it remains unclear whether they would have been eligible for ribociclib treatment in accordance with the NATALEE criteria (Figure 3).

Out of 2038 patients with HR+ HER2− eBC, 141 (6.9%) patients met the indication for all three agents (Figure 4). A total of 171 (8.4%) patients met the criteria for abemaciclib and ribociclib, and 5 (0.2%) could have been treated with olaparib or ribociclib. The largest subgroup, however, consisted of patients with an indication for ribociclib only (368, 18.1%). In total, 1353 (66.4%) of all patients with HR+ HER2− eBC did not meet the criteria for any form of combined oral maintenance therapy (Figure 5).

Therapy monitoring of oral endocrine-based maintenance therapies in early breast cancer is performed at regular intervals. The approval of ribociclib led to increased therapeutic options. However, therapy duration of ribociclib is one year longer compared to abemaciclib, which leads to a higher demand of health care providers’ personnel capacity (Figure 1). Table 2 describes the estimated duration of patient contacts during therapy management. Time requirement per patient contact will be reduced in the course of therapy. Approximately 31 patients per year will be eligible for abemaciclib treatment, whereas 69 patients per year will be eligible for abemaciclib or ribociclib treatment (Table 3 and Table 4). Provided 100% therapy adherence, health care providers will spend 134 h for therapy management of abemaciclib per year and 368 h for therapy management of ribociclib per year. Thus, the approval of ribociclib leads to a significantly higher demand of health care providers’ personnel capacity. Assuming a full-time position with 8 h/day for a year with 220 working days (weekdays minus public holidays and statutory vacation days in Germany), the 368 h for doctor/patient contacts in the context of potential ribociclib prescriptions represent a part-time position at 21% (0.21).

## 4. Discussion

To the best of our knowledge, this is the largest analysis on the potential indication for all three agents used for endocrine-based maintenance therapy in early breast cancer. While olaparib and abemaciclib are only recommended for patients with node-positive HR+ HER2− disease, the NATALEE study included node-negative patients with intermediate risk of relapse as well. Accordingly, we show that the number of potential candidates for combined adjuvant treatment will more than double due to ribociclib approval in 2024. We could furthermore demonstrate that the approval of ribociclib leads to three times higher demand of health care providers’ personnel capacity.

While 7% of all evaluated patients met the indications for all three agents, 18% were eligible for ribociclib only. It remains to be discussed how prescribing ribociclib for this large group of patients may impact the workload of outpatient oncologic clinics. Particularly before starting ribociclib therapy and during the first two cycles, blood sampling is recommended every 2 weeks, which may be reduced to monthly from the third cycle. From then on, laboratory examinations are only required if clinically indicated. In September 2024, the recommended number of electrocardiograms was reduced to two (before therapy initiation and 2 weeks afterwards) from three [21]. Despite this reduction in diagnostic ECG screenings, there remains a significantly increased number of additional patient appointments in clinics due to consultations, counseling sessions, and follow-up visits. Socioeconomic adjustments, such as expanding staffing capacity, seem both reasonable and necessary to manage this increased workload resulting from the higher frequency of doctor/patient interactions.

When comparing the results presented here with those from other real-world analyses, there are some minor differences in the percentage distribution of the respective patient cohorts for each substance. In contrast to another real-world analysis which found that a total of 43% of all HR+/HER2− eBC cases met the NATALEE inclusion criteria [20], only 34% of patients in our study met these criteria. However, among the T2 N0 G2 cohort, there remain 47 patients (2%) who did not receive genomic testing due to low-risk clinical-pathological characteristics and may also have been considered potential candidates for ribociclib therapy depending on their genomic risk levels.

The determination of the required ER and PR expression for a positive HR status at a minimum of 1% was based on the results of a large retrospective data analysis from the National Cancer Database (NCDB), which demonstrated an overall survival (OS) benefit in patients with low ER expression (1–10%) who received endocrine therapy compared to those who did not [22]. In this respect, the present analysis differs from a previous study by Schäffler et al., which defined a positive HR status as 10% or higher [20].

OlympiA and monarchE trials enrolled patients with high-risk tumors; however, inclusion criteria in the NATALEE study were much broader, thus allowing for inclusion of patients with intermediate risk profiles. As a result, the question arose whether all patients fulfilling the inclusion criteria should be offered combined therapy in the clinical routine. At the ESMO Congress 2024, Fasching et al. reported that ribociclib improved invasive disease-free survival (iDFS) in patients with node-negative tumors from 87.0% to 92.1% after 4 years (absolute benefit 5.1%, HR 0.666) [15]. While 88% of patients enrolled in the NATALEE study had node-positive tumors [4,15], 76% of potential candidates for ribociclib therapy in our study were node-positive. Thus, our real-world analysis revealed a higher percentage of node-negative patients who are eligible for therapy compared to the NATALEE study cohort. The study primarily included patients with high tumor stages; in the real-world context, the proportion of patients with higher tumor burden appears to be lower. Compared to other analyses [20], the proportion of patients with higher tumor burden (N+) generally appears to be lower in the real-world context than in the NATALEE study.

In our study, 15% of patients were identified as potential candidates for adjuvant abemaciclib. This is in accordance with other global real-world analyses showing that this population may comprise between 14% and 18% of all patients with HR+ HER2− tumors [20,23,24]. An important principle in the context of planned neoadjuvant treatment is the precise documentation of the number of sonographically suspicious axillary lymph nodes before starting therapy. This is crucial because the number of metastatic nodes after treatment can be lower due to treatment effects, and it is worth noting that the indication for abemaciclib therapy can be based on both pretherapeutic and post-surgery tumor stage, whichever meets the monarchE inclusion criteria [25,26].

When evaluating patients eligible for adjuvant olaparib, Dannehl et al. identified a total of 7.5% of all patients with HR+ HER2− eBC as potential candidates [27], which closely aligns with the 7.2% found in our study. However, it is important to note that the literature indicates that only 2.7% to 7.8% of all patients with HR+ HER2− eBC have a gBRCA mutation, with varying prevalence rates for gBRCA1 and gBRCA2 mutations [28,29]. Thus, a maximum of 12 out of 146 potential candidates for olaparib would actually be able to receive olaparib. Interestingly, some mutational carriers may be more likely to be eligible for olaparib if they receive primary surgery, since patients with ≥4 positive lymph nodes at the time of diagnosis do not always reach a CPS + EG score of 3 or higher.

It remains to be discussed whether all patients with an indication for at least one of the three substances for oral maintenance therapy would actually begin treatment in clinical practice, and whether they would complete the entire course of therapy. The latest study data indicate that 35.5% of patients discontinued ribociclib early, with a median time to discontinuation of 4 months. In the ribociclib plus non-steroidal aromatase inhibitor (NSAI arm), the primary reason for early discontinuation was adverse events (AEs), which occurred in 19.5% of cases [4,30,31]. Special situations, such as those involving older patients, as well as patient-related factors like therapy adherence and compliance, could reduce the actual number of patients undergoing oral maintenance therapy in everyday clinical settings, even though the prescribing information generally allows for dose adjustments or temporary therapy pauses in the case of severe adverse drug reactions [32]. Since combined endocrine therapy with ribociclib is prescribed for a fixed duration of 3 years, it is essential to consider the health-related quality of life (HR-QoL) in these patients. For ribociclib in eBC, there are only a few studies assessing HR-QoL [33] and overall, there is a general lack of real-world data on HR-QoL in patients treated with CDK 4/6 inhibitors [34,35]. Another aspect to consider is the potential forensic implications that could arise from the approval of ribociclib for HR2+ HER2− eBC, particularly if genomic testing is not conducted in patients with T2 N0 G2 disease. This could hypothetically lead to undertreatment and subsequent worsening of prognosis for these patients, potentially resulting in legal disputes.

## 5. Conclusions

This single-center analysis examined the number of potential candidates for combined endocrine therapy in a real-world context. While just under 7% of the cohort had an indication for all three agents (olaparib, abemaciclib, and ribociclib), over 18% met the approval criteria for ribociclib only, whereas over 66% of patients had no indication for combined endocrine therapy at all. To the best of our knowledge, this is the largest analysis addressing all three therapy strategies. It remains to be discussed to what extent the broad indication criteria for the NATALEE study will increase the workload in clinics through more frequent physician/patient interactions. It also remains open how therapy recommendations will impact the actual treatments administered, as more frequent patient visits and potential side effects may impact both compliance and therapy adherence.

## Figures and Tables

**Figure 1 cancers-17-00145-f001:**
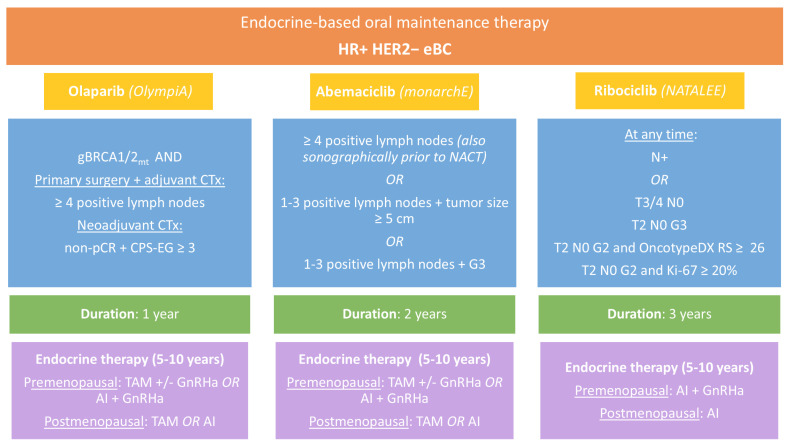
The most important characteristics of three available strategies for early HR+ HER2− eBC. Abbreviations: AI: aromatase inhibitor; gBRCA1/2mt: germline breast cancer 1/2 mutation; CTX: chemotherapy; GnRHa: gonadotropin-releasing hormone agonist; RS: recurrence score; TAM: tamoxifen.

**Figure 2 cancers-17-00145-f002:**
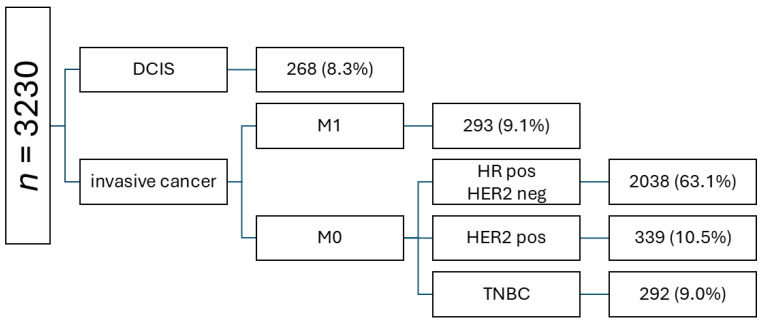
Flowchart of all patients with newly diagnosed breast cancer at the University Hospital Schleswig-Holstein, Campus Lübeck.

**Figure 3 cancers-17-00145-f003:**
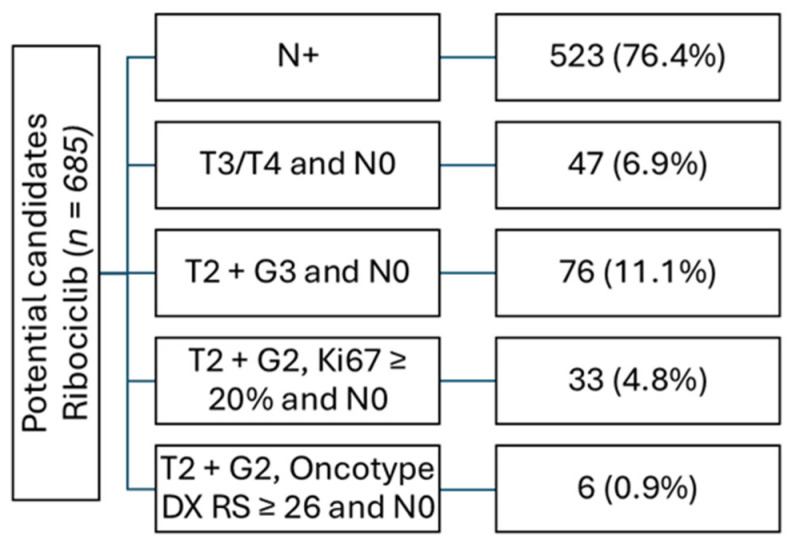
Potential candidates for ribociclib categorized according to the eligibility criteria of the NATALEE study.

**Figure 4 cancers-17-00145-f004:**
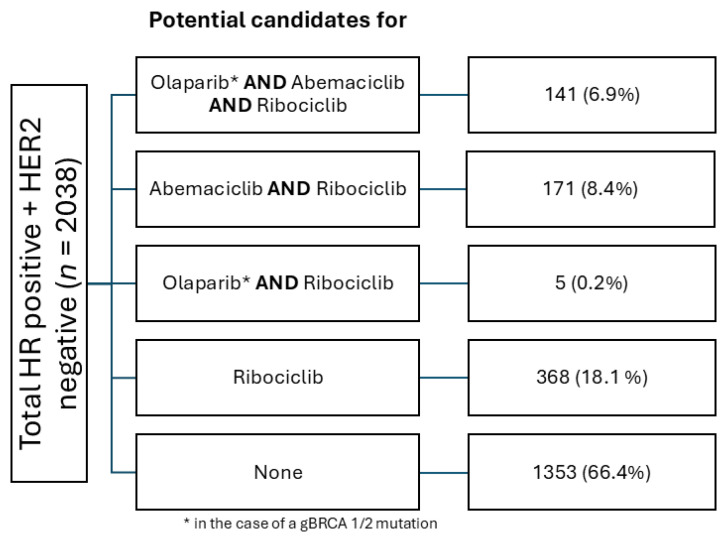
Potential candidates for oral endocrine-based maintenance therapy in early breast cancer.

**Figure 5 cancers-17-00145-f005:**
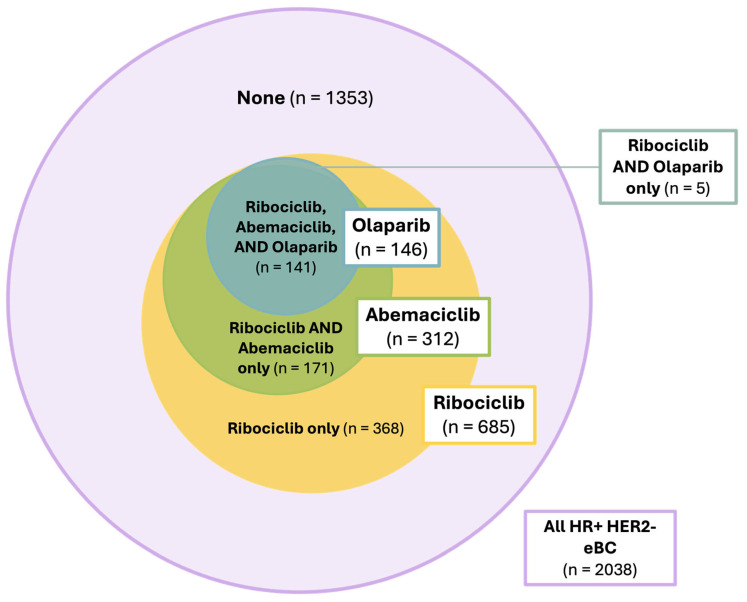
Venn diagram showing all patients with multiple and single indications for combined endocrine-based therapy in HR+ HER2− eBC. The pink circle represents all the patients in our analysis with HR+ HER2− early breast cancer (*n* = 2038). The yellow circle includes all patients indicated for ribociclib (*n* = 685) and the light-green circle represents all patients indicated for abemaciclib (*n* = 312). The small turquoise circle denotes all patients indicated for olaparib (*n* = 146).

**Table 1 cancers-17-00145-t001:** Characteristics of approval trials for endocrine-based maintenance therapy.

	Olaparib	Abemaciclib	Ribociclib
**Trial**	OlympiA [9,13]	monarchE [3,14]	NATALEE [4,15,16]
**Design**	Double-blind, placebo-controlled	Open-label	Open-label
**Dosing schedule**	300 mg BID	150 mg BID	400 mg QD for 3 weeks + 1 week off
**Inclusion criteria** ECOG-Status 0–1 ≥18 years old Completed definitive surgery	**HER2**−, **gBRCA1/2mt eBC** At least 6 cycles of CTx (anthracyclines, taxanes, or both) **AND** ^1^ Primary surgery + adjuvant CTx: ≥4 positive lymph nodes **OR** NACT: non-pCR + CPS + EG ≥ 3	**HR+ HER2**− **eBC** ≥4 positive lymph nodes (also sonographically prior to NACT) **OR** 1–3 positive lymph nodes + tumor size ≥ 5 cm **OR** 1–3 positive lymph nodes and G3 **OR** 1–3 positive lymph nodes and centrally assessed Ki-67 ≥ 20% (cohort 2; not included in the approval label)	**HR+ HER2**− **eBC** At any time: N+ **OR** T3/4 N0 **OR** T2 N0 G3 **OR** T2 N0 G2 and high-risk gene expression assay, e.g., OncotypeDX RS ≥ 26 **OR** T2 N0 G2 and Ki-67 ≥ 20%
**Number of patients**	1836	5637	5101
**Number of patients with HR+ HER2**− **eBC**	325 (17.7%)	5637 (100%)	5101 (100%)
**Treatment duration**	1 year	2 years	3 years
**Endocrine therapy**	Premenopausal: TAM +/− GnRHa **OR** AI + GnRHa Postmenopausal: TAM **OR** AI	Premenopausal: TAM +/− GnRHa **OR** AI + GnRHa Postmenopausal: TAM **OR** AI	Premenopausal: AI + GnRHa Postmenopausal: AI
**iDFS**	HR 0.628; 95% CI 0.504–0.779 (*overall population*)	HR 0.680; 95% CI 0.599–0.772; nominal *p* < 0.001	HR 0.715; 95% CI 0.609–0.840; *p* < 0.0001
**OS**	HR 0.678; 95% CI 0.503–0.907 *(overall population)*	Immature	Immature
**Approval**	FDA: March 2022 EMA: August 2022	FDA: March 2023 EMA: April 2022	FDA: September 2024 EMA: November 2024

^1^ Criteria for patients with HR+ HER2− disease. Abbreviations: AI: aromatase inhibitor; BID: lat. bis in die/eng. twice daily; CI: confidence interval; CTx: chemotherapy; CPS + EG: Combined Positive Score for Estrogen and HER2 receptors; ECOG: Eastern Cooperative Oncology Group; EMA: European Medicines Agency; FDA: Food and Drug Administration; gBRCA1/2mt: germline breast cancer 1/2 mutation; GnRHa: gonadotropin-releasing hormone agonist; HR: hazard ratio; iDFS: invasive disease-free survival, NACT: neoadjuvant chemotherapy; OS: overall survival; pCR: pathological complete response; QD: lat. quaque die/eng. once daily; RS: recurrence score; TAM: tamoxifen.

**Table 2 cancers-17-00145-t002:** Estimated duration of patient contacts of health care providers during the management of endocrine-based maintenance therapies in early breast cancer.

Therapy Week	Control Interval	Number of Patient Contacts	Time per Patient Contact	Total Time in Hours
1–8	2 weeks	4	20 min	1.33 (80 min)
9–52	4–8 weeks	8	15 min	2
53–104	8–12 weeks	6	10 min	1
105–156	8–12 weeks	6	10 min	1

**Table 3 cancers-17-00145-t003:** Estimated time of total patient contacts of health care providers during the management of abemaciclib.

Cohorts	Number of Patients	Total Time per Patient in hours	Time in Hours for all patients
Cohort year 1	31	3.33 (200 min)	103
Cohort year 2	31	1	31
**Total time per year for all patients (Cohort 1+2)**			134

**Table 4 cancers-17-00145-t004:** Estimated time of total patient contacts of health care providers during the management of ribociclib.

Cohorts	Number of Patients	Total Time per Patient in hours	Time in Hours for all Patients
Cohort year 1	69	3.33 (200 min)	230
Cohort year 2	69	1	69
Cohort year 3	69	1	69
**Total time per year for all patients (Cohorts 1–3)**			368

## Data Availability

The original contributions presented in this study are included in the article. Further inquiries can be directed to the corresponding author.

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
