# Peer review of "Oral Maintenance Therapy in Early Breast Cancer—How Many Patients Are Potential Candidates?"

_cancers, 2025, doi:10.3390/cancers17010145_

Round 1

Reviewer 1 Report

Comments and Suggestions for Authors

Oral maintenance therapy in early breast cancer - how many patients are potential candidates?

A brief summary:

This study evaluates the patient number that are considered as potential candidates for one or more maintenance therapy options at a relative large patient population size. Some findings from this study showed that the number of potential candidates for combined adjuvant treatment will more than double due to ribociclib approval in 2024. And the approval of ribociclib leads to three times higher demand of health care providers’ personnel capacity. However, it also showed that the demand of health care providers’ personnel capacity is higher in the therapy duration of ribociclib than abemaciclib. The population size of the study is relative large, however, it seems the authors didn’t clearly express their intention based on the data or insight finding to highlight the real purpose/ important (advantages and limitation) of this study. I would suggest to add the clarification of the purpose of the study in the introduction or discussion.

Here are some comments:

Line 27-29, maybe the two paragraphs in the simple summary can be combined into one.

Line 36-40, a total of 685 patients (33.6%), what is the total patients number here? Because referring to the total breast cancer cases (n=3230), 33.6% is not 685, please explain. In addition, the sentence “All of these patients met the criteria for ribociclib based on the NATALEE trial and 368 (18.1%) were eligible exclusively for ribociclib, including all node-negative patients.” Is not clear. Are all of patients referring to 523 patients or others? And 368 (18.1%) again, not sure how the percentage is calculated. Because a total of 685 patients (33.6%) qualified for one or more of the three agents, and 141 patients (6.9%) met the criteria for all three agents. It seems that 141 patients should belongs to the 685 patients, the percentage should be 20.6% from the total of 685 patients.  Please make the overall result displayed correctly and clarify the total patients in the study.

Line 67, there are lots of abbreviations in Table 1, please include their full names under the table.

Line 73, what is BRCA stands for? Line 130, what is TNM? Line 150, PN2 and PN3, please explain what they stand for? Line 248, 256,284, what are OS, iDFS, NSAI?

Line 152, what is CPS-EG score? Would you include more details about this score? How this score is used to initial clinical stage at diagnosis?

Line 193, figure 5 needs to add figure description to explain the cycles with different colors.

Line 213-215, for table 3 and 4, can those two table use the same format in calculating time in hours? Can all those number in minutes are changed to in hours for easily follow? Also include unites when descripting those number. In addition, the description in the tables are confusing. Such as Time in hours and total time per year in table 4, would it be better to change them to total time for all patients (hours) and total time for 3 treatment( because you are adding all hours from 3 year)?

Line 260-261, the finding from this study revealed a lower percentage of node-negative patients. What would be the major cause for the lower percentage of node-negative patients compared to the NATALEE study? Please explain. Other than lower percentage, what other advantages of this study to bring to the patient benefit?

Author Response

A brief summary:

This study evaluates the patient number that are considered as potential candidates for one or more maintenance therapy options at a relative large patient population size. Some findings from this study showed that the number of potential candidates for combined adjuvant treatment will more than double due to ribociclib approval in 2024. And the approval of ribociclib leads to three times higher demand of health care providers’ personnel capacity. However, it also showed that the demand of health care providers’ personnel capacity is higher in the therapy duration of ribociclib than abemaciclib. The population size of the study is relative large, however, it seems the authors didn’t clearly express their intention based on the data or insight finding to highlight the real purpose/ important (advantages and limitation) of this study. I would suggest to add the clarification of the purpose of the study in the introduction or discussion.

Here are some comments:

Line 27-29, maybe the two paragraphs in the simple summary can be combined into one.

Thank you for the structured feedback; we have connected the sentence accordingly.

Line 36-40, a total of 685 patients (33.6%), what is the total patients number here? Because referring to the total breast cancer cases (n=3230), 33.6% is not 685, please explain. In addition, the sentence “All of these patients met the criteria for ribociclib based on the NATALEE trial and 368 (18.1%) were eligible exclusively for ribociclib, including all node-negative patients.” Is not clear. Are all of patients referring to 523 patients or others? And 368 (18.1%) again, not sure how the percentage is calculated. Because a total of 685 patients (33.6%) qualified for one or more of the three agents, and 141 patients (6.9%) met the criteria for all three agents. It seems that 141 patients should belongs to the 685 patients, the percentage should be 20.6% from the total of 685 patients.  Please make the overall result displayed correctly and clarify the total patients in the study.

Thank you for the comment. We have moved the total number (2,038 Patients with HR+ HER2- ebC), to which the 33.6% refers, to the beginning of the sentence. We have revised the abstract and consistently stated the percentages in relation to the total number, ensuring greater clarity regarding the statistical values.

Line 67, there are lots of abbreviations in Table 1, please include their full names under the table.

We have listed all abbreviations as table captions and also added the abbreviations for figure 1.

Line 73, what is BRCA stands for? Line 130, what is TNM? Line 150, PN2 and PN3, please explain what they stand for? Line 248, 256,284, what are OS, iDFS, NSAI?

We have spelled out and introduced all abbreviations accordingly.

Line 152, what is CPS-EG score? Would you include more details about this score? How this score is used to initial clinical stage at diagnosis?

Thank you for the inquiry regarding the CPS-EG score. We have introduced the abbreviation: Combined Positive Score for Estrogen and HER2 receptors accordingly. A description of the meaning of this score was already provided in the introduction (lines 95-99).

Line 193, figure 5 needs to add figure description to explain the cycles with different colors.

Thank you for the suggestion. We have added the description and listed the individual colored circles separately.

Line 213-215, for table 3 and 4, can those two table use the same format in calculating time in hours? Can all those number in minutes are changed to in hours for easily follow? Also include unites when descripting those number. In addition, the description in the tables are confusing. Such as Time in hours and total time per year in table 4, would it be better to change them to total time for all patients (hours) and total time for 3 treatment( because you are adding all hours from 3 year)?

Thank you for the comments on the time-tables. We have now standardized the units and, for better clarity, included the minute count in parentheses for hour values with decimals. We also made a correction in Table 2 for weeks 53-156, as the number of patient contacts did not match the control intervals. Table 4 was adjusted for better presentation. In the case of a three-year treatment with ribociclib, three cohorts will be treated starting from year 3. Therefore, the hours for all three cohorts were added together and listed. We hope that the tables are now more clearly structured.

Line 260-261, the finding from this study revealed a lower percentage of node-negative patients. What would be the major cause for the lower percentage of node-negative patients compared to the NATALEE study? Please explain. Other than lower percentage, what other advantages of this study to bring to the patient benefit?

You have indeed identified a very important part of the text. The statement that our study included a smaller cohort of node-negative patients is unfortunately incorrect (inconsistent with the percentage figures mentioned in the previous sentence). We apologize for this initially incorrect sentence. On the contrary, the number of node-negative patients appears to be higher in the real-world context (as also observed in other analyses). In the NATALEE study, patients with a higher tumor burden were evidently included more frequently. We have therefore corrected this passage and expanded on it further.

Reviewer 2 Report

Comments and Suggestions for Authors

This manuscript describes the oral maintenance therapy in early breast cancer, specifically focusing on the use of PARP inhibitors and CDK4/6 inhibitors Olaparib, Abemaciclib, and Ribociclib for patients with hormone receptor-positive, HER2-negative early breast cancer. Olaparib, a PARP inhibitor, is approved for high-risk patients with germline BRCA1/2 mutations, while CDK4/6 inhibitors offer therapeutic potential for a broader patient population, as their effectiveness is independent of BRCA mutation status. CDK4/6 inhibitors are approved for use in the metastatic setting. The manuscript also examines the number of potential candidates for combined endocrine therapy in a real-world context. While just under 7% of the cohort qualified for all three agents (Olaparib, Abemaciclib, and Ribociclib), more than 18% met the approval criteria for Ribociclib alone, and over 66% of patients did not qualify for combined endocrine therapy. I recommend the publication of this important work.

1.    Authors did not include Palbociclib. It would be great if they could comment or describe about Palbo in the description of manuscript.

Author Response

This manuscript describes the oral maintenance therapy in early breast cancer, specifically focusing on the use of PARP inhibitors and CDK4/6 inhibitors Olaparib, Abemaciclib, and Ribociclib for patients with hormone receptor-positive, HER2-negative early breast cancer. Olaparib, a PARP inhibitor, is approved for high-risk patients with germline BRCA1/2 mutations, while CDK4/6 inhibitors offer therapeutic potential for a broader patient population, as their effectiveness is independent of BRCA mutation status. CDK4/6 inhibitors are approved for use in the metastatic setting. The manuscript also examines the number of potential candidates for combined endocrine therapy in a real-world context. While just under 7% of the cohort qualified for all three agents (Olaparib, Abemaciclib, and Ribociclib), more than 18% met the approval criteria for Ribociclib alone, and over 66% of patients did not qualify for combined endocrine therapy. I recommend the publication of this important work.

  1. Authors did not include Palbociclib. It would be great if they could comment or describe about Palbo in the description of manuscript.

Dear Author,

Thank you very much for the appreciative report and the effort you put into reviewing it. We are pleased that you recommend our manuscript for publication. In lines 102-104, the manuscript already states that, similar to the PALLAS and PENELOPE-B studies, Palbociclib has not shown a benefit for patients with early breast cancer (eBC). Therefore, there is no approval for Palbociclib in patients without metastases. Therefore, this therapeutic agent was not relevant for our analysis.